# Association of smoking with abdominal adipose deposition and muscle composition in Coronary Artery Risk Development in Young Adults (CARDIA) participants at mid-life: A population-based cohort study

James G. Terry[ID][1]*, Katherine G. Hartley[1], Lyn M. Steffen[2], Sangeeta Nair[1], Amy C. Alman[ID][3], Melissa F. Wellons[1], David R. Jacobs, Jr.[ID][2], Hilary A. Tindle[ID][1,4], John Jeffrey Carr[ID][1]

1 Vanderbilt University Medicine Center, Nashville, Tennessee, 2 University of Minnesota School of Public Health, Minneapolis, Minnesota, 3 University of South Florida, Tampa, Florida, 4 Geriatric Research Education and Clinical Centers (GRECC), Veterans Affairs Tennessee Valley Healthcare System, Nashville, Tennessee

* james.g.terry@vumc.org

**Data Availability Statement:** Data are available from the CARDIA Coordinating Center: http://www.

## Abstract

### Background

Smokers have lower risk of obesity, which some consider a "beneficial" side effect of smoking. However, some studies suggest that smoking is simultaneously associated with higher central adiposity and, more specifically, ectopic adipose deposition. Little is known about the association of smoking with intermuscular adipose tissue (IMAT), an ectopic adipose depot associated with cardiovascular disease (CVD) risk and a key determinant of muscle quality and function. We tested the hypothesis that smokers have higher abdominal IMAT and lower lean muscle quality than never smokers.

### Methods and findings

We measured abdominal muscle total, lean, and adipose volumes (in cubic centimeters) and attenuation (in Hounsfield units [HU]) along with subcutaneous (SAT) and visceral adipose tissue (VAT) volumes using computed tomography (CT) in 3,020 middle-aged Coronary Artery Risk Development in Young Adults (CARDIA) participants (age 42–58, 56.3% women, 52.6% white race) at the year 25 (Y25) visit. The longitudinal CARDIA study was initiated in 1985 with the recruitment of young adult participants (aged 18–30 years) equally balanced by female and male sex and black and white race at 4 field centers located in Birmingham, AL, Chicago, IL, Minneapolis, MN, and Oakland, CA. Multivariable linear models included potential confounders such as physical activity and dietary habits along with traditional CVD risk factors. Current smokers had lower BMI than never smokers. Nevertheless, in the fully adjusted multivariable model with potential confounders, including BMI and CVD risk factors, adjusted mean (95% CI) IMAT volume was 2.66 (2.55–2.76) cm$^3$ in current

cardia.dopm.uab.edu/contact-cardia. A description of the NHLBI policies governing the data and describing access to the data can be found at the following website: http://www.cardia.dopm.uab.edu/study-information/nhlbi-data-repository-data. Others have access to these data in the same manner as the authors, and the authors do not have any special access privileges that others would not have.

**Funding:** Funding: JGT, LMS, DRJ, and JJC and the CARDIA Study as a whole are supported through NHLBI awards to University of Alabama at Birmingham (HHSN268201800005I & HHSN268201800007I), Northwestern University (HHSN268201800003I), University of Minnesota (HHSN268201800006I), and Kaiser Foundation Research Institute (HHSN268201800004I), and Vanderbilt School of Medicine (R01-HL098445) and NIA an Intramural Research Program of the National Institute on Aging (NIA) and an intra-agency agreement between NIA and NHLBI (AG0005).The funders had no role in study design, data collection and analysis, decision to publish, or preparation of the manuscript.

**Competing interests:** The authors have declared that no competing interests exist.

**Abbreviations:** AGES-Reykjavik, Age, Gene/Environment Susceptibility-Reykjavik study; CAC, coronary artery calcification; CARDIA, Coronary Artery Risk Development in Young Adults; CRP, C-reactive protein; CT, computed tomography; CVD, cardiovascular disease; DXA, dual-energy X-ray absorptiometry; FF, fast food; HbA1c, hemoglobin A1c; Health ABC, Health, Aging, and Body Composition study; HU, Hounsfield units; IL-6, interleukin-6; IMAT, intermuscular adipose tissue; MIPAV, Medical Image Processing, Analysis, and Visualization; SAT, subcutaneous adipose tissue; SDH, succinate dehydrogenase; SSB, sugar-sweetened beverage; STROBE, Strengthening the Reporting of Observational Studies in Epidemiology; TNF-α, tumor necrosis factor-α; VAT, visceral adipose tissue; Y25, year 25.

smokers ($n = 524$), 2.36 (2.29–2.43) cm$^3$ in former smokers ($n = 944$), and 2.23 (2.18–2.29) cm$^3$ in never smokers ($n = 1,552$) ($p = 0.007$ for comparison of former versus never smoker, and $p < 0.001$ for comparison of current smoker versus never and former smoker). Moreover, compared to participants who never smoked throughout life (41.6 [41.3–41.9] HU), current smokers (40.4 [39.9–40.9] HU) and former smokers (40.8 [40.5–41.2] HU) had lower lean muscle attenuation suggesting lower muscle quality in the fully adjusted model ($p < 0.001$ for comparison of never smokers with either of the other two strata). Among participants who had ever smoked, pack-years of smoking exposure were directly associated with IMAT volume (β [95% CI]: 0.017 [0.010–0.025]) ($p < 0.001$). Despite having less SAT, current smokers also had higher VAT/SAT ratio than never smokers. These findings must be viewed with caution as residual confounding and/or reverse causation may contribute to these associations.

## Conclusions

We found that, compared to those who never smoked, current and former smokers had abdominal muscle composition that was higher in adipose tissue volume, a finding consistent with higher CVD risk and age-related physical deconditioning. These findings challenge the belief that smoking-associated weight loss or maintenance confers a health benefit.

## Author summary

### Why was this study done?

- Smoking and obesity are, separately, well-known health risks for cancer and cardiovascular disease (CVD).

- Smokers often have lower risk of obesity measured using BMI, leading to the misconception of a "beneficial side effect" to smoking.

- Even with a lower BMI, smokers may have a higher risk of depositing fat (more properly called adipose tissue) in and around organs and tissues compared to those who never smoked. This type of fat carries higher risk and may interfere with organ and tissue functions.

### What did the researchers do and find?

- We used computed tomography (CT) to measure abdominal fat deposited just below the skin's surface (subcutaneous fat), around organs including the intestines (visceral fat) and abdominal muscles (intermuscular fat), and inside the muscles (intramuscular fat) in 3,020 middle-aged participants in the Coronary Artery Risk Development in Young Adults (CARDIA) study.

- We found that current smokers had higher proportions of fat within their abdominal muscles and visceral fat around their internal organs compared to never smokers, whereas those who had quit smoking had intermediate levels of visceral and intramuscular fat.

**What do these findings mean?**

- Despite lower BMI and subcutaneous fat, smokers appear to be at risk of accumulating organ-associated fat and intramuscular fat that have been shown to increase circulating blood fats and sugar.

- This may, in turn, explain some of the hidden, higher risk of CVD and disability in smokers.

## Introduction

Historically, cigarette advertisements promoted the idea that a lower body weight was a possible beneficial side effect of smoking [1,2]. It is, therefore, not surprising that smokers cite concern about weight gain as a barrier to smoking cessation [3,4]. Indeed, BMI is lower in current smokers compared to nonsmokers, and those who quit smoking tend to gain weight [5–8]. However, smoking has also been associated with central fat patterning [9,10]. Central fat patterning suggests higher ectopic fat deposition within or around non-adipose tissues or organs (e.g., liver, muscle, or heart), which is, in turn, strongly associated with diabetes, cardiovascular disease (CVD) risk, and all-cause mortality [11–19].

Studies using computed tomography (CT) to delineate specific adipose depots suggest that smoking is associated with higher visceral adipose tissue (VAT) [20–23], but less is known about the role of smoking in ectopic adipose depots other than VAT. Ectopic adipose deposition within skeletal muscles (intermuscular adipose tissue [IMAT]) is an independent risk factor for coronary artery calcification (CAC), diabetes, and cardiovascular and all-cause death [16,18,19]. Accumulation of IMAT and associated changes in muscle composition and function are suspected contributors to disability in the elderly [24]. The prevalence of severe, clinically significant loss of lean muscle mass, known as sarcopenia, ranges from 5% to 10% in community-dwelling populations over age 65 to perhaps 30% in those over age 80 [24]. Further, the Women's Health Initiative dual-energy X-ray absorptiometry (DXA) substudy found that 17% of participants had sarcopenic obesity [25]. Risk factors likely contributing to both ectopic adipose tissue deposition and muscle deconditioning include sedentary lifestyle [26,27], smoking [26,27], and diets rich in fast food (FF), sugar-sweetened beverages (SSBs), and alcohol [23,28–30], and smoking.

We evaluated the role of smoking history in abdominal adipose deposition and, specifically, its role in muscle composition and quality using CT in more than 3,000 Coronary Artery Risk Development in Young Adults (CARDIA) participants aged 43 to 55. Our overarching hypothesis is that, even at mid-life, current smokers have higher IMAT and poorer muscle quality than former or never smokers after adjusting for generalized obesity and potentially confounding factors such as diet quality, alcohol consumption, and physical activity.

## Materials and methods

### Study population

The CARDIA study began in 1985 with recruitment of 5,115 participants aged 18 to 30 years at field centers located in Birmingham, AL, Chicago, IL, Minneapolis, MN, and Oakland, CA [31]. Recruitment was balanced for equal inclusion of black and white and female and male participants, age (18–24, 25–30 years), and education (≤12 years, >12 years). The current

study includes data from participants who agreed to undergo abdominal CT scan at the year 25 (Y25) examination. A total of 3,499 participants were examined in clinic at Y25, representing 72% of the original cohort. Of these 3,499, 3,172 underwent abdominal CT, and 327 were excluded due to weight, inability to fit in the CT scanner, or (rarely) pregnancy. Of 3,172 participants with abdominal CT scans, 3,020 participants (95.2% of those who underwent CT) had complete measures of abdominal adipose tissues and muscle composition along with smoking status and key covariables including BMI. The manuscript proposal and analysis plan for the present study was approved by the CARDIA Publications and Presentations Committee on March 14, 2018, and assigned #A-1811 (S1 CARDIA Proposal). The analysis protocol was followed as approved, and as with all CARDIA manuscripts, the manuscript underwent data confirmation and CARDIA peer review prior to study approval. This study is reported as per the Strengthening the Reporting of Observational Studies in Epidemiology (STROBE) guideline (S2). All participants provided written informed consent, and institutional review boards from each field center and the coordinating center approved the study annually.

## Clinical evaluations

**Clinical measures.** Clinic visit procedures were standardized and consistent across examinations as previously published in detail [31]. Blood pressure was measured in triplicate after a 5-minute rest using an automated blood pressure monitor (Omron model HEM907XL; Omron Healthcare Inc., Lake Forest, IL) with the average of the second and third measurements used in analyses. Fasting plasma lipids and lipoproteins were measured using enzymatic methods at Northwest Lipids Research Laboratory (Seattle, WA). Serum glucose was measured using the Roche Modular P hexokinase method (Roche Diagnostics, Rotkreuz, Switzerland), and percent hemoglobin A1c (HbA1c) was measured using Tosoh G7 HPLC (Tosoh, San Francisco, CA). Plasma C-reactive protein (CRP) was measured using high-sensitivity nephelometry-based method (BNII nephelometer, Dade Behring, Eschborn, Germany). Diabetes was defined as fasting glucose $\geq$7.0 mmol/L ($\geq$126 mg/dL), self-report of oral hypoglycemic medications or insulin, 2-hour postload glucose $\geq$11.1 mmol/L ($\geq$200 mg/dL), or HbA1c $\geq$6.5% at the Y25 visit.

Use of antihypertensive and lipid-lowering treatments was collected through interviewer-administered questionnaires. At the baseline visit, when participants were 18 to 30 years of age, diabetes and use of antihypertensive and lipid-lowering medicines was extremely rare, so these measures were not included in any models adjusted for baseline exposures.

## Lifestyle measures

**Smoking history.** Based on interviewer-administered questionnaires, cigarette smoking was classified as never, former, or current at each CARDIA visit (S1 Form, S2 Form). For the cross-sectional analyses at Y25, smoking data from each participant visit were reviewed, and participants were coded as follows: (1) never smokers only if they always reported never smoking and never reported past smoking at any visit; (2) former smokers if they denied current smoking but had admitted current smoking or past smoking at any other CARDIA visit; or (3) current smokers if the participant admitted current smoking at the Y25 visit. For those who reported past or current smoking at any CARDIA visit, the age at which the participant became a regular smoker was queried. The average number of cigarettes smoked per day across all CARDIA visits attended from baseline (Y0) through Y25 was calculated and divided by 20 cigarettes per package to determine mean packs per day. To reliably estimate the smoking exposure during the 25-year CARDIA follow-up, mean packs per day was multiplied by the number of years between consecutive visits in which the participant reported currently

smoking. Thus, a participant who reported being a former smoker at baseline (age 18–30) and was never a current smoker during follow-up was coded as having 0 pack-years, while a participant who was a current smoker at all attended visits through Y25 and averaged smoking 20 cigarettes per day would be coded as having 25 pack-years' exposure. Serum cotinine was measured only at the CARDIA baseline visit as an independent marker of active current smoking [32–35]. Participants with cotinine ≥14 ng/mL were coded positive for active smoking based on research establishing that cut-off as providing >98% accuracy in smoking status classification [32–35].

## Other lifestyle and clinical measures

Alcohol consumption was computed as milliliters per day based upon self-reported drinks per week of beer (12-ounce glass), wine (5-ounce glass), and liquor (1.5-ounce shot). Self-reported usual weekly intake of FF and SSBs were used as surrogates for diet quality. FF intake was categorized as never consumes, 1–2 visits per week, or ≥3 meals per week. SSB was categorized as never consumes, 1–2 SSBs, or ≥3 SSBs per week. The CARDIA Physical Activity History questionnaire was used to estimate weekly leisure, occupational, and household physical exertion over the past 12 months [36]. Education was self-reported in years, and maximum years of education was included in models. Weight and height were measured with participants wearing light clothing and no shoes. Body weight was measured to the nearest 0.2 kg on a calibrated scale, whereas height was measured to the nearest 0.5 cm using a fixed vertical ruler. BMI was calculated as weight in kilograms divided by height in meters squared.

## CT measures of arterial calcification and adipose deposition

Participants underwent a multidetector CT chest and abdomen scans using a standardized protocol [15,16,37–39]. The scans were performed at CARDIA field centers using 64-channel multidetector GE CT scanners (GE Healthcare Milwaukee, WI) at the Birmingham, AL, and Oakland, CA, centers and Siemens CT scanners (Siemens, Erlangen, Germany) at the Chicago, IL, and Minneapolis, MN, centers. ECG gating was used for the cardiac scan, and a quality control phantom was included (INTableTM Calibration Pad, Image Analysis, Columbia, KY). CT images were electronically transmitted to the central CT reading center located at Wake Forest University School of Medicine, Winston-Salem, NC. Experienced image analysts measured calcified plaque using FDA-approved workstations (Aquarius Workstation, TeraRecon, Foster City, CA) producing total calcium scores based on the Agatston method [38,40]. In the present study, CAC was defined as present for scores >0 Agatston units. Abdominal CT scans were analyzed using the 50-cm display field of view. Adipose tissue depots were measured volumetrically within a 10-mm block of 10 × 1 mm or 8 × 1.25 mm contiguous slices based on the nominal slice thickness produced by the specific scanner centered at the level of the disk between the 4th and 5th lumbar vertebrae as previously described [16–18]. Tissues with attenuation of −190 through −30 Hounsfield units (HU) were defined as adipose tissue. Medical Image Processing, Analysis, and Visualization (MIPAV) software was used to quantify subcutaneous (SAT) and VAT volume.

Abdominal muscle composition (fat, lean, and total) was measured volumetrically from a 10-mm block of contiguous slices centered between the 3rd and 4th lumbar disks [16,18]. The abdominal muscles were measured at the L3–L4 level to avoid changes in muscle orientation related to the pelvic bones in some individuals at the L4–L5 level. Muscle volumes at L3–L4 and L4–L5 were highly correlated ranging from 0.87 (rectus) to 0.98 (psoas and paraspinous). Pixels within muscle with attenuation of −190 to −30 HU were defined as adipose tissue and −29 to 160 HU as lean tissue. Fat, lean, and total muscle volumes were quantified for the psoas,

paraspinous, lateral oblique, and rectus muscles using a custom MIPAV plug-in developed by study investigators [16,18]. Measures of left and right muscles in each group were highly correlated, so mean adipose, lean, and total volumes of the left and right sides were calculated and analyzed for all abdominal muscles. Mean muscle attenuation within the range of −29 to 160 HU was calculated across all muscles and considered a measure of intramuscular adipose content.

Analysis reliability of CT measures was assessed through blinded intra- and inter-reader re-reads of 158 scan pairs (approximately 5%). Overall (intra- and inter-reader) technical error in re-analysis of 158 pairs of scans was 6.6% for CAC, 6.0% for VAT, and 7.7% for psoas muscle total volume with correlations for re-reads >0.95 in each measure.

## Statistics

Abdominal adipose tissue and muscle composition outcome variables were analyzed as continuous data. Smoking status (never, former, current) at Y25 and pack-years of smoking exposure were the primary independent variables. Model covariables were chosen a priori. Model 1 included age, sex, race, and field center, with model 2 adding education, alcohol consumption, FF consumption frequency, SSB consumption, physical activity, and BMI. Model 3 included model 2 covariables plus diabetes status, systolic blood pressure, CRP, triglycerides, use of anti-hypertensive and cholesterol medicines, and prevalent CAC. VAT/SAT ratio was added to model 3 for muscle outcomes to test associations after adjustment for central adipose deposition. Analyses were repeated with baseline smoking status and baseline cotinine as predictors in models including the above Y25 covariables (model 3) or baseline covariables age, education, height, physical activity, alcohol consumption, FF consumption, systolic blood pressure, fasting triglycerides and glucose, and BMI along with sex, race, and field center.

We tested two-way and three-way sex, race, and Y25 smoking status interactions in fully adjusted models predicting the primary variables of interest, IMAT volume, muscle attenuations, and IMAT/lean volume ratio. There were no significant interactions for sex, race, and smoking status (all interactions >0.15). All analyses were performed using STATA version 15.

## Results

At the Y25 visit, just over half of the participants were never smokers ($n$ = 1,552), 31% were former smokers ($n$ = 944), and 17% were current smokers ($n$ = 524) (Table 1). Compared to never smokers, current smokers were less likely to be female or white, had fewer years of education, were less physically active, had higher alcohol intake, and were more likely to consume SSBs and FF. Current smokers had lower BMI but higher blood pressure, triglycerides, and CRP compared to never smokers. Of note, baseline BMI did not differ by smoking status at CARDIA baseline (24.5 [4.8], 24.5 [4.8], and 24.4 [4.8] kg/m$^2$ among never, former, and current smokers, respectively, $p$ = 0.95). Current smokers were much more likely to have prevalent CAC than either never smokers or former smokers.

### Associations of smoking history at CARDIA Y25 visit with abdominal adipose tissues and muscle composition

CT measures of abdominal adipose tissue and muscle composition are shown in Table 2. Unadjusted SAT volume was >25 cm$^3$ lower in current smokers than either former or never smokers. VAT volume was nominally higher in former smokers compared to never smokers. Compared to never smokers, VAT/SAT ratio was higher in former smokers and higher still in current smokers. Total and lean muscle volumes were slightly higher in current smokers compared to former smokers. IMAT volume and IMAT/lean ratio were each markedly higher in

**Table 1. Participant characteristics at Y25 visit according to Y25 smoking status[*].**

| Characteristic | All (N = 3,020) | Smoking Status | | | p[†] |
|---|---|---|---|---|---|
| | | Never (n = 1,552) | Former (n = 944) | Current (n = 524) | |
| *Demographic* | | | | | |
| Age, y | 50.1 (3.6) | 49.9 (3.6) | 50.8 (3.5) | 49.6 (3.7) | <0.001 |
| Female, % | 56.3% | 56.4% | 58.9% | 51.2 | 0.02 |
| White, % | 52.6% | 53.0% | 60.1% | 38.0 | <0.001 |
| Education, y | 15.6 (2.6) | 16.1 (2.4) | 15.5 (2.6) | 13.9 (2.2) | <0.001 |
| *Lifestyle* | | | | | |
| Age started smoking, y[‡] | 18.0 (4.6) | n/a | 17.8 (3.6) | 18.4 (5.7) | 0.02 |
| 25-year smoking exposure, pack-years[‡] | 2.3 (0.1–10.0) | n/a | 0.4 (0.0–2.9) | 12.1 (5.7–19.6) | <0.001 |
| Physical activity, units | 277 (126–486) | 271 (124–490) | 312 (144–504) | 227 (105–413) | <0.001 |
| Alcohol intake, mL/d | 2.4 (0–14.7) | 0.0 (0.0–10.0) | 4.8 (0.0–17.0) | 8.9 (0.0–28.6) | <0.001 |
| SSB, % | | | | | |
| Never consumes | 33.9% | 35.2% | 37.8% | 23.3% | |
| <1–2/wk | 43.3% | 44.5% | 43.3% | 39.7% | |
| ≥3/wk | 22.8% | 20.4% | 18.9% | 37.0% | <0.001 |
| FF, % | | | | | |
| Never consumes | 33.3% | 32.3% | 39.2% | 26.0% | |
| <1–2 meals/wk | 37.4% | 36.5% | 36.1% | 42.2% | |
| ≥3 meals/wk | 29.3% | 31.2% | 24.7% | 31.9% | <0.001 |
| BMI, kg/m$^2$ | 30.2 (7.1) | 30.6 (7.1) | 30.2 (7.4) | 29.2 (6.4) | <0.001 |
| WC, cm | 94.6 (15.8) | 94.7 (15.7) | 94.6 (16.3) | 94.3 (14.7) | 0.88 |
| *Clinical* | | | | | |
| Diastolic BP, mmHg | 75.0 (11.2) | 74.5 (11.0) | 74.7 (11.0) | 77.1 (11.5) | <0.001 |
| Systolic BP, mmHg | 119.8 (16.0) | 119.1 (15.5) | 119.3 (16.2) | 122.9 (16.9) | <0.001 |
| HTN treatment, % | 27.4% | 25.5% | 28.0% | 32.1% | 0.01 |
| Cholesterol treatment, % | 15.8% | 15.0% | 17.0% | 16.0% | 0.43 |
| Total cholesterol, mg/dL | 192.5 (37.0) | 192.0 (35.8) | 192.8 (36.5) | 193.3 (40.5) | 0.76 |
| LDL cholesterol, mg/dL | 112.0 (32.8) | 113.3 (31.8) | 110.6 (32.5) | 111.0 (35.4) | 0.10 |
| HDL cholesterol, mg/dL | 58.0 (18.0) | 57.4 (16.9) | 59.5 (18.9) | 57.0 (19.6) | 0.006 |
| Triglycerides, mg/dL | 93 (68–134) | 90 (65–129) | 93 (68–134) | 100 (75–157) | <0.001 |
| CRP, mg/dL | 1.4 (0.6–3.5) | 1.4 (0.6–3.3) | 1.3 (0.6–3.3) | 1.8 (0.8–4.3) | <0.001 |
| Diabetes, % | 12.4% | 11.3% | 12.5% | 15.3% | 0.06 |
| CAC prevalence, % | 28.2% | 23.7% | 28.5% | 41.0% | <0.001 |

[*]Smoking history based on participant reported status at Y25 and confirmed across each CARDIA visit attended from baseline through Y25.

[†]Based on ANOVA (continuous variables), Kruskal-Wallis (continuous, non-normally distributed variables), or chi-squared (categorical variables).

[‡]Available for 943 former smokers and 523 current smokers (1,466 total).

**Abbreviations:** BP, blood pressure; CAC, coronary artery calcification; CARDIA, Coronary Artery Risk Development in Young Adults; CRP, C-reactive protein; FF, fast food; HDL, high-density lipoprotein; HTN, hypertension; LDL, low-density lipoprotein; n/a, not applicable; SSB, sugar-sweetened beverage; WC, waist circumference; Y25, year 25

current and former smokers compared to never smokers. Lean muscle attenuation was lower in former or current smokers than never smokers (each comparison $p < 0.0001$), suggesting an association of smoking with higher intramyocellular fat.

Multivariable models for the association of Y25 smoking status with adipose tissues and muscle composition are shown in Table 3. SAT was approximately 11% lower in current smokers than either former or never smokers in the minimally adjusted model 1, but these

**Table 2. Unadjusted Y25 CT measures of abdominal adipose tissues and muscle composition [mean (SD)] by Y25 smoking status*.**

| Abdominal Measure | | All (N = 3,020) | Smoking Status | | | | | |
|---|---|---|---|---|---|---|---|---|
| | | | Never (n = 1,552) | Former (n = 944) | Current (n = 524) | $p_{former}$ vs. never | $p_{current\ vs.\ former}$ | $p_{current\ vs.\ never}$ |
| **Adipose Depots** | SAT | 334.9 (169.1) | 344.4 (170.0) | 333.9 (170.7) | 308.8 (161.0) | 0.13 | 0.006 | <0.001 |
| | VAT | 132.0 (73.8) | 130.1 (73.0) | 136.0 (76.0) | 130.8 (71.9) | 0.05 | 0.20 | 0.85 |
| | VAT/SAT ratio | 0.456 (0.311) | 0.431 (0.254) | 0.469 (0.304) | 0.505 (0.443) | 0.003 | 0.034 | <0.001 |
| **Muscle Composition** | Total volume | 20.4 (5.2) | 20.4 (5.3) | 20.3 (5.1) | 20.9 (5.0) | 0.42 | 0.029 | 0.09 |
| | Lean volume | 18.0 (4.6) | 18.1 (4.8) | 17.7 (4.5) | 18.2 (4.4) | 0.07 | 0.041 | 0.48 |
| | IMAT volume | 2.35 (1.62) | 2.25 (1.53) | 2.42 (1.64) | 2.51 (1.82) | 0.013 | 0.28 | 0.001 |
| | IMAT/lean ratio | 0.136 (0.098) | 0.129 (0.092) | 0.141 (0.102) | 0.143 (0.105) | 0.003 | 0.80 | 0.007 |
| | Attenuation | 41.1 (6.4) | 41.7 (6.4) | 40.4 (6.2) | 40.6 (6.4) | <0.001 | 0.61 | <0.001 |

*Smoking status based on participant reported status at Y25 and confirmed across each CARDIA visit attended from baseline through Y25; comparisons based on ANOVA; tissue volumes are in cm$^3$, and attenuation is in HU.

**Abbreviations:** CARDIA, Coronary Artery Risk Development in Young Adults; CT, computed tomography; HU, Hounsfield units; IMAT, intermuscular adipose tissue; SAT, subcutaneous adipose tissue; VAT, visceral adipose tissue; Y25, year 25

differences were attenuated in models 2 and 3. Former and current smokers had significantly higher VAT compared to never smokers in model 2 only. VAT/SAT ratio was higher in current smokers compared to never smokers in all models. Also shown in Table 3, total muscle volume was higher by 0.50 cm$^3$ among current smokers compared to never smokers in the fully adjusted model (model 4). Current smokers had IMAT volume that was 0.30 cm$^3$ (approximately 11%) higher than former smokers and 0.43 cm$^3$ (approximately 16%) higher than never smokers in the fully adjusted model (model 4). Lean muscle attenuation was lower in current smokers or former smokers compared to those who never smoked in all models ($p < 0.0001$ for all comparisons).

Among those with a history of smoking, we also tested the association of pack-years of smoking exposure during CARDIA follow-up with adipose volumes and muscle composition. As shown in Fig 1A, pack-years were positively associated with IMAT volume ($p < 0.001$) in fully adjusted models. For reference, 10 pack-years' higher smoking exposure was associated with an additional 0.17 cm$^3$ IMAT (approximately 7% higher volume per 10 pack-years based on mean 2.45 cm$^3$ IMAT among ever smokers). Fig 1B and Fig 1C demonstrate the association of pack-years' smoking exposure with higher IMAT/lean volume ratio ($p < 0.001$) and lower muscle attenuation ($p = 0.015$). There were no significant associations of pack-years of

**Table 3. Multivariable models of Y25 CT abdominal adipose tissues and muscle composition (least squares mean [95% CI]) by Y25 smoking status*.**

| Abdominal Depot | Measure | Model | Smoking Status | | | $p_{former\ vs.\ never}$ | $p_{current\ vs.\ former}$ | $p_{current\ vs.\ never}$ |
|---|---|---|---|---|---|---|---|---|
| | | | Never | Former | Current | | | |
| **Adipose Depots** | SAT | 1 | 344.3 (336.7–352.0) | 338.5 (328.4–348.3) | 301.6 (287.7–314.4) | 0.36 | <0.001 | <0.001 |
| | | 2 | 336.8 (332.9–340.6) | 334.3 (329.4–339.1) | 330.7 (323.9–337.6) | 0.43 | 0.41 | 0.14 |
| | | 3 | 335.8 (332.0–339.7) | 334.30 (329.4–339.1) | 333.5 (326.6–340.4) | 0.61 | 0.86 | 0.57 |
| | VAT | 1 | 130.5 (127.1–134.0) | 134.5 (130.0–139.0) | 132.2 (126.0–138.2) | 0.18 | 0.54 | 0.66 |
| | | 2 | 128.8 (126.1–131.4) | 133.7 (130.4–137.0) | 138.8 (134.1–143.5) | 0.022 | 0.08 | <0.001 |
| | | 3 | 130.3 (127.9–132.8) | 133.5 (130.4–136.6) | 134.4 (130.0–138.9) | 0.12 | 0.74 | 0.13 |
| | VAT/SAT ratio | 1 | 0.432 (0.419–0.445) | 0.461 (0.445–0.478) | 0.514 (0.492–0.536) | 0.007 | <0.001 | <0.001 |
| | | 2 | 0.438 (0.425–0.451) | 0.463 (0.447–0.480) | 0.494 (0.470–0.517) | 0.021 | 0.039 | <0.001 |
| | | 3 | 0.444 (0.431–0.457) | 0.463 (0.447–0.479) | 0.477 (0.453–0.500) | 0.08 | 0.36 | 0.021 |

*(Continued)*

**Table 3.** (Continued)

| Abdominal Depot | Measure | Model | Smoking Status | | | $p_{\text{former vs. never}}$ | $p_{\text{current vs. former}}$ | $p_{\text{current vs. never}}$ |
|---|---|---|---|---|---|---|---|---|
| | | | Never | Former | Current | | | |
| Muscle Composition | Total volume | 1 | 20.4 (20.3–20.6) | 20.6 (20.4–20.8) | 20.2 (19.9–20.5) | 0.25 | 0.04 | 0.20 |
| | | 2 | 20.3 (20.2–20.4) | 20.5 (20.3–20.7) | 20.8 (20.6–21.1) | 0.09 | 0.022 | <0.001 |
| | | 3 | 20.3 (20.2–20.5) | 20.5 (20.3–20.6) | 20.8 (20.5–21.0) | 0.14 | 0.06 | 0.002 |
| | | 4 | 20.3 (20.2–20.5) | 20.5 (20.3–20.6) | 20.8 (20.5–21.0) | 0.16 | 0.06 | 0.003 |
| | Lean volume | 1 | 18.1 (17.9–18.2) | 18.1 (17.9–18.3) | 17.6 (17.4–17.8) | 0.85 | 0.001 | <0.001 |
| | | 2 | 18.0 (17.9–18.1) | 18.0 (17.9–18.2) | 18.0 (17.8–18.2) | 0.70 | 0.88 | 0.90 |
| | | 3 | 18.0 (17.9–18.1) | 18.0 (17.9–18.2) | 17.9 (17.7–18.2) | 0.85 | 0.72 | 0.81 |
| | | 4 | 18.0 (17.9–18.1) | 18.0 (17.9–18.2) | 18.0 (17.7–18.2) | 0.86 | 0.71 | 0.81 |
| | IMAT volume | 1 | 2.26 (2.18–2.34) | 2.40 (2.30–2.50) | 2.51 (2.37–2.65) | 0.035 | 0.22 | 0.002 |
| | | 2 | 2.22 (2.16–2.28) | 2.36 (2.29–2.43) | 2.69 (2.59–2.79) | 0.003 | <0.001 | <0.001 |
| | | 3 | 2.23 (2.17–2.29) | 2.36 (2.29–2.43) | 2.66 (2.56–2.77) | 0.005 | <0.001 | <0.001 |
| | | 4 | 2.23 (2.18–2.29) | 2.36 (2.29–2.43) | 2.66 (2.55–2.76) | 0.007 | <0.001 | <0.001 |
| | IMAT/lean ratio | 1 | 0.130 (0.125–0.132) | 0.138 (0.130–0.142) | 0.148 (0.139–0.155) | 0.042 | 0.068 | <0.001 |
| | | 2 | 0.129 (0.125–0.132) | 0.136 (0.131–0.141) | 0.155 (0.148–0.161) | 0.014 | <0.001 | <0.001 |
| | | 3 | 0.129 (0.125–0.133) | 0.136 (0.132–0.141) | 0.154 (0.147–0.160) | 0.015 | <0.001 | <0.001 |
| | | 4 | 0.129 (0.126–0.133) | 0.136 (0.132–0.141) | 0.153 (0.147–0.160) | 0.021 | <0.001 | <0.001 |
| | Attenuation | 1 | 41.7 (41.4–42.0) | 40.7 (40.4–41.1) | 40.1 (39.7–40.7) | <0.001 | 0.07 | <0.001 |
| | | 2 | 41.7 (41.4–41.9) | 40.8 (40.5–41.2) | 40.2 (39.7–40.7) | <0.001 | 0.06 | <0.001 |
| | | 3 | 41.6 (41.3–41.9) | 40.8 (40.4–41.2) | 40.3 (39.8–40.8) | <0.001 | 0.14 | <0.001 |
| | | 4 | 41.6 (41.3–41.9) | 40.8 (40.4–41.2) | 40.4 (39.9,40.9) | <0.001 | 0.16 | <0.001 |

*Smoking status based on participant reported status at Y25 and confirmed across each CARDIA visit attended from baseline through Y25; comparisons based on ANOVA; tissue volumes are in $cm^3$, and attenuation is in HU. Model 1: age, race, sex, and center; model 2: model 1 + education, physical activity, alcohol consumption, SSB consumption, FF consumption, and BMI; model 3: model 2 + diabetes, cholesterol treatment, hypertension treatment, systolic BP, triglycerides, CRP, and prevalent CAC; model 4: model 3 + VAT/SAT ratio (muscle composition measures only); tissue volumes are in $cm^3$, and attenuation is in HU.

**Abbreviations:** BP, blood pressure; CAC, coronary artery calcification; CARDIA, Coronary Artery Risk Development in Young Adults; CRP, C-reactive protein; CT, computed tomography; FF, fast food; HU, Hounsfield units; SAT, subcutaneous adipose tissue; SSB, sugar-sweetened beverage; VAT, visceral adipose tissue; Y25, year 25

exposure with abdominal adipose or total or lean muscle volumes in fully adjusted models. In former smokers, we also tested the association between years since quitting and muscle composition and abdominal adipose measures and found no significant associations.

## Associations of smoking history at CARDIA baseline visit with Y25 abdominal adipose tissue and muscle composition

Associations of baseline smoking status and cotinine levels with Y25 adipose and muscle composition are shown in Supporting Information S1 and S2 Tables, respectively. Participants who were smokers at baseline (age 18–30 years) had higher Y25 VAT/SAT ratio than never smokers ($p = 0.005$) after adjusting for either concurrent baseline or Y25 factors (S1 Table). Compared to those who had never smoked at baseline, current smokers had 0.27 $cm^3$ higher IMAT volume 25 years later along with lower muscle attenuation in all models. Participants with baseline cotinine $\geq$14 ng/mL, suggestive of current smoking, had higher VAT/SAT ratio 25 years later after adjustment for baseline or Y25 covariables (S2 Table). IMAT volume was approximately 0.2 $cm^3$ higher and muscle attenuation was lower (each $p < 0.001$) in those with high cotinine.

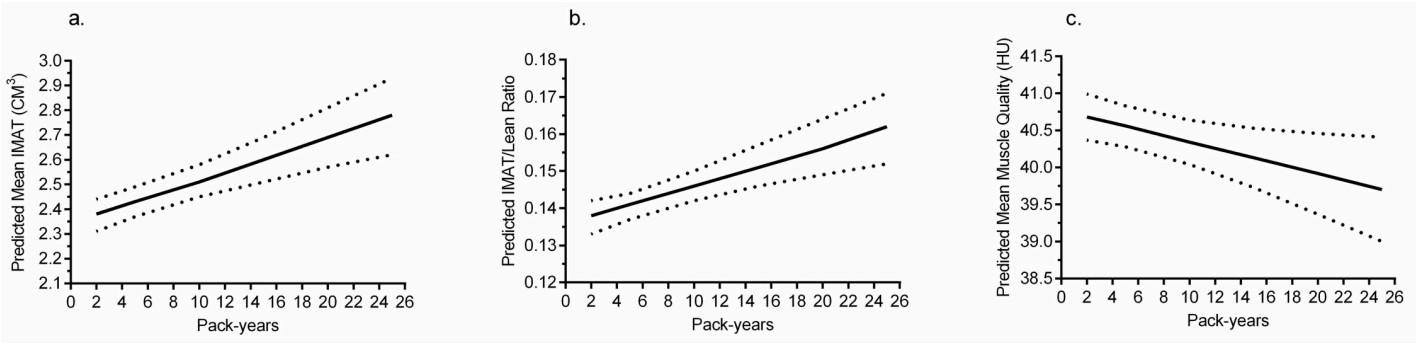

**Fig 1. Predicted IMAT volume (cm$^3$), IMAT/lean volume ratio, and lean muscle quality (HU) based on pack-years smoked from CARDIA Y0 through Y25.** (a) Predicted IMAT, (b) IMAT/lean ratio, and (c) lean muscle quality are plotted against continuous pack-years during CARDIA adjusted for age, race, sex, center, education, physical activity, alcohol consumption, SSB consumption, FF consumption, BMI, diabetes, cholesterol treatment, hypertension treatment, systolic BP, triglycerides, CRP, prevalent CAC, and VAT/SAT ratio. Pack-years was a significant predictor of IMAT volume (β [95% CI] 0.017 [0.010–0.025], $p < 0.001$), IMAT/lean volume ratio (β [95% CI] 0.001 [0.0006–0.0015], $p < 0.001$), and lean muscle quality (β [95% CI] −0.043 (−0.077 to −0.008], $p = 0.016$). BP, blood pressure; CAC, coronary artery calcification; CARDIA, Coronary Artery Risk Development in Young Adults; CRP, C-reactive protein; FF, fast food; HU, Hounsfield units; IMAT, intermuscular adipose tissue; SAT, subcutaneous adipose tissue; SSB, sugar-sweetened beverage; VAT, visceral adipose tissue; Y0, baseline; Y25, year 25.

## Discussion

In this multicenter study of healthy middle-aged participants followed 25 years, we found that, compared to never smokers, current smokers had higher abdominal muscle adipose volume and lower lean muscle attenuation, suggesting lower muscle quality. The adverse association of smoking with muscle composition persisted after adjustment for other lifestyle factors and both BMI and visceral fat. The frequent longitudinal assessment of smoking status (mean 7.4 ± 1.1 assessments over 25 years) provided rigorous, comprehensive capture of cumulative smoking exposure throughout early to middle adult life. These data suggest that the lower BMI associated with smoking masks higher ectopic adipose tissue deposition and lower muscle quality.

Smoking is associated with lower risk of obesity, and smoking cessation is thought to contribute to weight gain [5–8]. Moreover, the inverse association between smoking and weight has been shown to strengthen with age [41], a finding replicated in CARDIA in that baseline BMI was not associated with smoking status but, 25 years later, current smokers had markedly lower BMI compared to either never or former smokers. However, a recent mendelian randomization study found that genes associated with higher BMI and waist circumference are associated with higher odds of being a current smoker and more intense smoking habits [42]. Thus, the direction of the association between smoking and measures of obesity could plausibly run in either direction. Regardless, the association of smoking with higher waist circumference is consistent with cross-sectional studies suggesting that smoking is associated with higher abdominal VAT [20,21,43]. Accumulation of VAT is strongly associated with poor CVD risk factor profiles and prevalent and incident CVD [11]. Though BMI was lower in current compared to never smokers, current smokers in CARDIA had higher VAT and VAT/ SAT ratio, an index of central fat deposition, after adjustment for possible confounders and BMI. Although smoking has previously been linked to higher central fat deposition, our findings suggest that smoking is adversely associated with muscle composition with poorer muscle quality apparent as early as mid-life. Current smokers had higher IMAT volume and higher proportion of IMAT relative to lean muscle compared to either never smokers or former smokers, findings that were not explained by lower BMI in smokers. Perhaps as importantly, current and former smokers in CARDIA had significantly lower lean muscle attenuation

compared to never smokers, suggesting higher intramyocellular adipose deposition and poorer quality lean muscle.

A recent meta-analysis estimates the excess risk for clinically significant sarcopenia associated with ever smoking at approximately 12%, although smoking-related risk was most apparent in elderly participants [44]. Extreme sarcopenic changes associated with functional decline and frailty would be expected to be rare at mid-life given that lean muscle mass peaks in early adulthood around age 30 and declines thereafter by about 1% per year in men [45]. However, a large study found that smoking and higher pack-years of exposure were associated with lower skeletal muscle mass among 845 healthy men aged 45 to 85 (mean age 64) [26]. In the present CARDIA analysis, lean muscle volume did not differ by smoking status. Total muscle volume was slightly higher in current smokers compared to never smokers, though quantitatively the difference was consistent with the amount of excess IMAT in smokers. Our data suggest that assessing only lean or total muscle volume not only may lead to potentially missing important associations between muscle composition and risk factors such as smoking but also could underestimate the pathological impact of IMAT accumulation [16,18,19,46,47].

IMAT accumulation is likely to contribute to decline in muscle function and mobility in older adults [48]. The Health, Aging, and Body Composition (Health ABC) study followed more than 1,600 healthy septuagenarians and found that thigh IMAT increased over 5 years even in those who lost weight [49]. A recent analysis from the Age, Gene/Environment Susceptibility (AGES)-Reykjavik Study reported that current smoking was associated with lower thigh muscle attenuation (suggesting fat infiltration) in septuagenarian men and women [50]. Further, higher pack-years of smoking was associated with lower muscle attenuation and peak muscle torque in women, but not men [50]. Taken together, the AGES-Reykjavik and CARDIA data suggest that smoking is associated with higher adipose tissue and poorer skeletal muscle quality from middle age through late in life. Increasing IMAT is associated with lower gait speed, grip strength, and other functional assessments that, in turn, portend falls and limited mobility with aging [48]. Indeed, in the Women's Health Initiative DXA study, low lean body mass—especially when combined with obesity—was strongly associated with risk of falls [25].

IMAT accumulation is associated with diabetes risk as shown in CARDIA and other studies [18,51,52]. Despite its association with lower risk of BMI-assessed obesity, current smoking is associated with higher risk of diabetes [51,52]. Diabetes, in turn, contributes to risk of sarcopenia in the elderly [53]. As demonstrated in the present study, the association of current smoking and smoking exposure with higher IMAT suggests a plausible mechanism through which smoking might pose a risk for diabetes. The 0.4 $cm^3$ difference in IMAT volume between current and never smokers is approximately 25% of the 1.6 $cm^3$ SD for IMAT among all participants. In context, we have previously shown that a full 1-SD higher IMAT level is associated with approximately 90% higher diabetes prevalence [18]. Excess accumulation of IMAT impairs glucose disposal via muscle insulin resistance, interrupting glucose metabolism in skeletal muscles which account for approximately 80% of glucose utilization [54]. Though smoking is associated with diabetes risk, to date, neither the American Diabetes Association nor the International Diabetes Foundation include current smoking in risk calculators for type 2 diabetes.

Mechanisms potentially linking smoking to muscle compositional changes and deconditioning are numerous. Muscle biopsies taken from current, long-term smokers had muscle fiber cross-sectional area 25% smaller than nonsmokers, suggesting an association of smoking with muscle wasting [55–57]. These observational data in humans are consistent with laboratory experiments showing that direct exposure to cigarette smoke reduced muscle fiber cross-sectional area and increased succinate dehydrogenase (SDH) activity in rats [55,58]. Muscle fiber

atrophy and higher SDH, a measure of oxidative activity, are likely adaptations to local hypoxia induced by cigarette smoke [55,58]. As a major source of reactive oxygen and nitrogen species, mainstream cigarette smoke likely foments oxidative stress and chronic inflammation, which could promote detrimental muscle morphologic and metabolic changes [59–60]. Though the present study showed higher IMAT and lower lean muscle quality rather than lean muscle loss per se, our findings stem from middle-aged participants at low risk of clinically significant sarcopenia. In light of the present CARDIA data, findings from an experimental mouse model fed a high-fat diet and given injections of nicotine or saline are of interest [61]. After 10 weeks on the high-fat diet, the nicotine-injected mice weighed less than controls but accumulated intra-myocellular lipid and had higher oxidative stress [61]. IMAT accumulation is also associated with pro-inflammatory circulating cytokines, including interleukin-6 (IL-6), CRP, and tumor necrosis factor-$\alpha$ (TNF-$\alpha$) [51]. In the presence of obesity, chronic oxidative and inflammatory stress are implicated in many disease processes, including sarcopenia, diabetes, and CVD [62].

## Limitations

Though smoking history was verified by cotinine measurement at baseline, smoking was thereafter self-reported, potentially resulting in misclassifying of risk. The CT adipose tissue and muscle composition outcome variables were obtained 25 years after the CARDIA baseline visit. Therefore, we quantified the association of smoking exposure assessed at multiple time points with abdominal adipose deposition measured at a single time point and so the present findings should be confirmed with longitudinal measurement of tissue changes. As such, we cannot definitively rule out reverse causality as a contributor to these associations. Two plausible scenarios are that cigarette smoking increases organ-related fat or that cigarette smoking is started in part because of a tendency towards fatness but is effective only in reducing non–organ-related fat. Although we adjusted for dietary and other habitual factors that may explain variation in muscle composition, it is possible that residual confounding exists.

## Conclusions

In the present study, IMAT volume was higher and muscle quality lower in current smokers compared to never smokers. Among ever smokers, having greater pack-years of exposure was directly associated with IMAT and inversely associated with muscle quality. Importantly, former smokers had muscle composition and quality intermediate between never smokers and current smokers, suggesting that cessation is worthwhile despite possible weight gain. These findings are important given the widespread misconception that smoking confers weight-related health benefits.

## Supporting information

**S1 STROBE checklist. STROBE, Strengthening the Reporting of Observational Studies in Epidemiology.**
(DOCX)

**S1 Table. Multivariable models of Y25 muscle composition (least squares mean [95% CI]) by baseline smoking status.** Y25, year 25.
(DOCX)

**S2 Table. Multivariable models of Y25 muscle composition (least squares mean [95% CI]) by baseline cotinine level.** Y25, year 25.
(DOCX)

**S1 CARDIA Proposal. CARDIA study publications and presentations form detailing the study.** CARDIA, Coronary Artery Risk Development in Young Adults.
(DOCX)

**S1 Form. CARDIA study smoking questionnaire administered at each clinic visit.** CARDIA, Coronary Artery Risk Development in Young Adults.
(PDF)

**S2 Form. CARDIA study smoking questionnaire administered at each clinic visit.** CARDIA, Coronary Artery Risk Development in Young Adults.
(PDF)

## Acknowledgments

We thank the investigators, the staff, and the participants of the Coronary Artery Risk Development in Young Adults (CARDIA) study for their dedication and highly valued contributions. This article has been reviewed by CARDIA for scientific content.

## Author Contributions

**Conceptualization:** James G. Terry, Katherine G. Hartley, Lyn M. Steffen, Sangeeta Nair, Melissa F. Wellons, David R. Jacobs, Jr., John Jeffrey Carr.

**Data curation:** James G. Terry, David R. Jacobs, Jr., John Jeffrey Carr.

**Formal analysis:** James G. Terry, David R. Jacobs, Jr.

**Funding acquisition:** John Jeffrey Carr.

**Investigation:** James G. Terry, Lyn M. Steffen, Sangeeta Nair, David R. Jacobs, Jr., John Jeffrey Carr.

**Methodology:** James G. Terry, Sangeeta Nair, Amy C. Alman, Melissa F. Wellons, David R. Jacobs, Jr., Hilary A. Tindle, John Jeffrey Carr.

**Project administration:** James G. Terry, John Jeffrey Carr.

**Resources:** John Jeffrey Carr.

**Software:** John Jeffrey Carr.

**Supervision:** David R. Jacobs, Jr., John Jeffrey Carr.

**Validation:** David R. Jacobs, Jr.

**Writing – original draft:** James G. Terry.

**Writing – review & editing:** James G. Terry, Katherine G. Hartley, Lyn M. Steffen, Sangeeta Nair, Amy C. Alman, Melissa F. Wellons, David R. Jacobs, Jr., Hilary A. Tindle, John Jeffrey Carr.

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
