## [Decision Letter · Decision Letter 0]

17 May 2020

Dear Dr. Terry,

Thank you very much for submitting your manuscript "Smoking is associated with abdominal adipose deposition and muscle composition in mid-life: The CARDIA Study" (PMEDICINE-D-19-04435) for consideration at PLOS Medicine. 

[LINK]

In light of these reviews, I am afraid that we will not be able to accept the manuscript for publication in the journal in its current form, but we would like to consider a revised version that addresses the reviewers' and editors' comments. Obviously we cannot make any decision about publication until we have seen the revised manuscript and your response, and we plan to seek re-review by one or more of the reviewers. 

We expect to receive your revised manuscript by Jun 01 2020 11:59PM - we are suggesting a slightly shortened timeline for resubmission given this paper is for the special issue (Determinants, Consequences and Management of Obesity), and that the revisions suggested by reviewers are not substantial. Please email us (plosmedicine@plos.org) if you have any questions or concerns.

We look forward to receiving your revised manuscript. 

Sincerely,

Emma Veitch, PhD

PLOS Medicine

On behalf of:

Adya Misra, PhD

Senior Editor 

PLOS Medicine

plosmedicine.org

*Please revise your title according to PLOS Medicine's style. Your title must be nondeclarative and not a question. It should begin with main concept if possible. "Effect of" should be used only if causality can be inferred, i.e., for an RCT. Please place the study design ("A randomized controlled trial," "A retrospective study," "A modelling study," etc.) in the subtitle (ie, after a colon) - in this case presumably "prospective cohort" would be appropriate.

*In the last sentence of the Abstract Methods and Findings section, please describe the main limitation(s) of the study's methodology - in the main discussion section the possibility of residual confounding is mentioned, but reverse causality (as noted by one reviewer) might also be a potential source of bias, and this could be discussed/mentioned as appropriate in either the main Discussion section or abstract, as appropriate.

*At this stage, we ask that you include a short, non-technical Author Summary of your research to make findings accessible to a wide audience that includes both scientists and non-scientists. The Author Summary should immediately follow the Abstract in your revised manuscript. This text is subject to editorial change and should be distinct from the scientific abstract. Please see our author guidelines for more information: https://journals.plos.org/plosmedicine/s/revising-your-manuscript#loc-author-summary

*We'd ask you to modify the referencing format to match PLOS Medicine's style (this should be numbered callouts in square brackets (eg [1, 2]) rather than superscript numerals. If referencing software was used this should be fairly quick and straightforward.

*Did your study have a prospective protocol or analysis plan? Please state this (either way) early in the Methods section.

*In the Abstract, currently effect estimates for differences between compared groups are presented, along with p-values. The journal normally expects that 95% confidence intervals (CI's) will also be given so that readers can understand the imprecision associated with the effects seen in the study. We'd ask that these are given both in the abstract and in the main text along with the main effect estimates.

*As the paper reports findings from a prospective cohort, we'd suggest that the study is reported according to the STROBE guideline; please include the completed STROBE checklist as Supporting Information. Please add the following statement, or similar, to the Methods: "This study is reported as per the Strengthening the Reporting of Observational Studies in Epidemiology (STROBE) guideline (SChecklist)." The STROBE guideline can be found here: http://www.equator-network.org/reporting-guidelines/strobe/. When completing the checklist, please use section and paragraph numbers, rather than page numbers.

*As noted by one reviewer, residual confounding (and/or reverse causation) are possible explanations for the effects seen in the analysis, which limit interpretation of the associations. We'd caution the authors to ensure that the writeup in the paper is not overly conclusive about a causal basis to the associations reported in the analysis - these should be presented as associations/links rather than definitive effects. 

Comments from the reviewers:

Reviewer #1: I confine my remarks to statistical aspects of this paper. These were well done and I recommend publication

Peter Flom

Reviewer #2: Terry and co-authors examined the association between cigarette smoking and intermuscular adipose tissue and attenuation in the well-described CARDIA population. Although the authors used rather unique measures in a large sample, I have a number of concerns specifically focused on the clinical/medical translation of the findings. 

1) The nature of the study is observational only, so no causality can be inferred on the basis of this data.

2) The analyses are done cross-sectional meaning that the direction of the effect cannot be explained. Previous studies (e.g., Mendelian Randomization studies) showed that individuals with obesity have a higher propensity to be a current smoker. 

3) The clinical relevance of the findings is poorly examined in the present study. For example, there is an approximate 0.4 units difference in IMAT volume between never and current smokers. The impact of this difference on health (e.g., diabetes or CAD risk) is not discussed. A mediation analysis would be helpful to see whether the association between smoking and health outcomes is mediated by the differences observed in the present study. Especially because the authors also showed a lower BMI in the current smokers, this would be interesting. Combining all anthropometric difference in current smokers would provide a clearer image to the impact on diseases like T2D (or Hb1Ac or HOMA-IR, in case they measured this).

4) It would have been interesting to examine the association of years of smoking cessation and the study outcomes. It seems that past smokers have an IMAT/Attenuation which is a bit in the middle between past en never smokers, but this group is very heterogenous in terms of the duration of smoking cessation. This would give this study a somewhat longitudinal aspect. 

5) It is unclear to this reviewer what the authors tried to do with the adjustment for the baseline variables (25 years ago!). 

Reviewer #3: This is an excellent paper on an important topic. I am surprised the analysis has not been done before. The authors have been thorough in their statistical analysis and the discussion is well-written.

I only have very tiny amendments:

-"type 2 diabetes" rather than "type ii diabetes." (the latter term is older nomenclature)

-In the opening sentence of the introduction, change "was" to "is." Alternatively, add the word "historically" to the beginning of the sentence to make the tenses match.

[LINK]

---

## [Decision Letter · Decision Letter 1]

8 Jun 2020

Dear Dr. Terry,

Thank you very much for re-submitting your manuscript "Smoking is associated with abdominal adipose deposition and muscle composition in CARDIA Study participants at mid-life: a prospective cohort study" (PMEDICINE-D-19-04435R1) for review by PLOS Medicine.

I have discussed the paper with my colleagues and the academic editor and it was also seen again by xxx reviewers. I am pleased to say that provided the remaining editorial and production issues are dealt with we are planning to accept the paper for publication in the journal.

[LINK]

We look forward to receiving the revised manuscript by Jun 15 2020 11:59PM. 

Sincerely,

Adya Misra, PhD

Senior Editor 

PLOS Medicine

plosmedicine.org

Requests from Editors:

Abstract

Could you please provide additional participant demographics? For instance, where the participants are from/where the registry is based

For p-values throughout, please use the format 0.xxx 

Please clarify this sentence “compared to never smokers, current and former smokers had abdominal muscles higher in adipose tissue”

Throughout- please provide a space between text and reference square brackets

Introduction

I would suggest an alternative for “svelte figure” and “quitters” 

I would suggest rephrasing “obese, ageing society” as it could be interpreted as stigmatising 

Could you please provide citations to or copies of the questionnaires used in the study 

Line 269- please provide the exact p-value here and also the data. PLOS do not permit instances of data not shown. The same goes for lines 317, 319 and 339

For all tables, please define CRP, CAC, HU in the footnotes

At line 371, please revisit the possibly contentious statement about the "tobacco industry's longstanding advertising campaign". The cited article appears to refer to adverts from the 1930s. We suggest deleting the relevant sentence.

Similarly, please ensure that the statement at lines 101-102 accurately reflects the content of the cited references. 

At line 471, unless there are surveys documenting the beliefs of "millions of smokers", please amend the wording.

Please quote exact p values or "p<0.001".

Please add full access details to reference 1.

Please adapt the title so that it is non-declarative in style, e.g., "Smoking and abdominal adipose deposition ...".

Please remove the word "prospective" from the title (we view this as a retrospective analysis of a prospectively gathered dataset).

Please remove all iterations of "[Internet]" from the reference list. 

Comments from Reviewers:

Reviewer #2: The authors addressed my concerns

[LINK]

---

## [Editor Report · Decision Letter 2]

18 Jun 2020

Dear Mr Terry, 

On behalf of my colleagues and the academic editor, Dr. Sanjay Basu, I am delighted to inform you that your manuscript entitled "Association of smoking with abdominal adipose deposition and muscle composition in Coronary Artery Risk Development in Young Adults (CARDIA) participants at mid-life: a population-based cohort study" (PMEDICINE-D-19-04435R2) has been accepted for publication in PLOS Medicine. 

PRODUCTION PROCESS

PRESS

PROFILE INFORMATION

Thank you again for submitting the manuscript to PLOS Medicine. We look forward to publishing it. 

Best wishes, 

Adya Misra, PhD

Senior Editor 

PLOS Medicine

plosmedicine.org